# SARS-CoV-2: Reinfection after 18 Months of a Previous Case with Multiple Negative Nasopharyngeal Swab Tests and Positive Fecal Molecular Test

**DOI:** 10.3390/medicina58050642

**Published:** 2022-05-06

**Authors:** Carlo Brogna, Barbara Brogna, Domenico Rocco Bisaccia, Marino Giuliano, Luigi Montano, Simone Cristoni, Mauro Petrillo, Marina Piscopo

**Affiliations:** 1Department of Research, Craniomed Group Facility SRL, 83038 Montemiletto, Italy; roccobisa@gmail.com; 2Department of Radiology, Moscati Hospital, Contrada Amoretta, 83100 Avellino, Italy; 3Marsanconsulting Srl Public Health Company, Via dei Fiorentini, 80133 Napoli, Italy; marino@marsanconsulting.it; 4Andrology Unit and Service of LifeStyle Medicine in Uro-Andrology, Local Health Authority (ASL), 84124 Salerno, Italy; luigimontano@gmail.com; 5ISB—Ion Source & Biotechnologies SRL, 20091 Bresso, Italy; simone.cristoni@gmail.com; 6Seidor Italy SRL, 21029 Milan, Italy; mauro.petrillo@seidor.com; 7Department of Biology, University of Naples Federico II, 80126 Napoli, Italy; marina.piscopo@unina.it

**Keywords:** COVID-19, SARS-CoV-2, qRT-PCR, feces, oropharyngeal swab, nasopharyngeal swab, faecal swab, viral pneumonia, chest, computed tomography

## Abstract

This short communication describes the reinfection after nearly 18 months of the same patient who was previously infected with coronavirus disease 2019 (COVID-19) and who showed multiple negative real-time quantitative reverse transcriptase-polymerase chain reaction (RT-qPCR) results by nasal swabs for severe acute respiratory syndrome coronavirus (SARS-CoV-2) but positive results on a fecal sample. We previously noted how, in the presence of symptoms suggestive of pneumonia, visible on a chest computed tomography (CT) scan and confirmed by fecal molecular testing, it was possible to draw the diagnosis of SARS-CoV-2 infection. One year later, the same patient was again affected by SARS-CoV-2. This time, the first antigenic nasal swab showed readily positive results. However, the patient’s clinical course appeared to be more attenuated, showing no signs of pulmonary involvement in the radiographic examinations performed. This case shows a novelty in the pulmonary radiological evaluation of new SARS-CoV-2 infection.

## 1. Introduction

More than two years have passed since the 2019 coronavirus disease pandemic (COVID-19), and the possibility of fecal-oral transmission continues to be under investigation.

Several researches pointed out how the fecal-oral transmission could be considered on par with airborne transmission [1,2,3,4,5,6,7,8,9,10,11,12,13,14,15].

On the other hand, many investigators have emphasized the importance of angiotensin-converting enzyme 2 (ACE2) and transmembrane serine protease 2 (TMPRSS2) receptors throughout the intestinal tract and how the virus can also replicate in the intestinal cells [16,17,18]. Some studies highlight this close connection between clinical manifestations and gastrointestinal or respiratory infection [19,20,21]. One of these is the case report of an 80-year-old man who was vaccinated and had gastrointestinal manifestations typical of severe acute respiratory syndrome coronavirus infection (SARS-CoV-2) but was nasopharyngeal molecular negative on tests obtained by real-time quantitative reverse transcriptase-polymerase chain reaction (qRT-PCR). However, he transmitted the virus to his roommate, who was positive to a nasopharyngeal qRT-PCR molecular swab, during his hospital stay before he died. Only after the elderly man’s subsequent autopsy was the virus found in many anatomic districts except the lung and the ophthalmic bulb [22]. Reinfection with SARS-CoV-2 in previously cured subjects is a phenomenon that is observed in particular with the last B.1.1.529 (Omicron) variant [23,24,25]. 

Furthermore, observational studies show that newly emerging variants of the virus could affect both vaccinated individuals and those with a previous SARS-CoV-2 infection [26,27,28], with milder symptoms for the latter, although data in the literature are discordant [27]. In this brief supplement of previously published work [29], we report for the first time a difference in the detection of viral RNA in the same patient and how the pulmonary aspect radiologically differs between the first and second infections.

We previously noted that viral RNA was found in the fecal swab despite multiple negative nasopharyngeal molecular tests. Almost 18 months after the first episode, the same unvaccinated subject was reinfected. However, molecular testing from a nasopharyngeal swab revealed a viral load this time. In addition, while COVID-19-typical pneumonia was visible at chest CT in the first episode, after 18 months it showed a normal lung that was not affected by the second infection.

## 2. A Brief Description of the Case

The 44-year-old patient, Caucasian, with arterial hypertension disease, chronically taking ace-inhibitors, a non-smoker, from a good socio-economic condition, had his first Covid-19 infection about 18 months earlier. The last time he came to our observation, he had severe dyspnea, desaturation, and heart palpitations. The first time, we found positive SARS-CoV-2 molecular tests in his feces, despite six negative nasopharyngeal molecular swabs [29]. At the end of the first infection, which occurred in autumn 2020, he had 16.44 AU/mL SARS-CoV-2 IgG and 59.50 AU/mL SARS-CoV-2 IgM (Abbott: AdviseDx SARS-CoV-2 IgG II assay—chemiluminescent microparticle immunoassay (CMIA)). At the end of January 2022 (i.e., almost 18 months after the infection reported in [29]), he was reinfected by SARS-CoV-2 and he was not vaccinated for SARS-CoV-2.

The patient’s symptoms were fever (37.5 °C), severe sore throat, nausea, cough, and general malaise. The patient also had thoracic pain. The first positive nasal antigenic swab was followed by confirmation of a positive molecular nasopharyngeal swab (Copan lot 2127827) with RT-PCR analysis (Viasure real-time PCR detection kit-Cortest Biotec). The oxygen saturation (SaO2) ranged from 94% to 96% for the period of illness (seven days). In addition, his wife and children became ill one more time, as in the first episode of autumn 2020. However, the overall symptoms were milder than the first time. 

In line with previous observations [30,31,32], the oral therapy adopted was amoxicillin with clavulanic acid 3 g/day for three days and 2 g/day for another three days, associated with azithromycin 500 mg/day for six days. As supplements, he took vitamin C (2 g/day for seven days), vitamin D (25,000 I.U. once a week), and probiotics (*Lactobacillus reuteri* 10 billion/day for 15 days) [33]. Blood tests (summarized in Table 1) revealed no concerns.

A chest computed tomography (CT) scan was requested due to the persistence of the thoracic pain. However, on the chest CT, the patient did not show any lung alterations (Figure 1).

On day ten, the patient was nasopharyngeal molecular swab negative (Copan lot 2127827). The patient tested negative on day 10, and the clinical state of discomfort lasted eight days in total. Currently, the patient has no symptoms, and his antibody titer is 25,563.3 AU/mL SARS-CoV-2 IgG (Abbott).

Moreover, the patient has not shown any symptoms attributable to a post-COVID-19 status.

## 3. Discussion

Cases of reinfection are usually reported in healthcare workers, and the rate of reinfection in the general population is underestimated [34]. The SARS-CoV-2 reinfection of the previously described case [29] showed milder symptoms. In the first infection, despite the multiple nasal swabs negativity for SARS-CoV-2 with fecal test positivity, the patient also showed signs of COVID-19 pneumonia on CT with IgM positivity on serology. However, the patient did not show any lung involvement during the second infection.

This case is in line with other reports [35,36] and studies [26,27,34,37], which show that reinfected cases have a more attenuated and less severe clinical course than the first time. 

Real-time quantitative reverse transcriptase-polymerase chain reaction (RT-qPCR) is the gold-standard laboratory technique for the identification of SARS-CoV-2 in the clinical setting, and nasopharyngeal swabs are considered the primary sample for COVID-19 tests with a sensitivity of around 70–80% [38,39,40]. However, false negative results are also possible [38,41], and other specimens, including fecal viral testing that has a high specificity to detect SARSCoV-2 despite a low sensitivity ranging from 37% to 60%, have also been considered [42,43]. However, some studies demonstrated that the clearance time of COVID-19 in the digestive tract was later than that in the respiratory tract [44,45]. Recently, Wu et al. [15] found that fecal positivity for SARS-CoV-2 was independent from the manifestation of gastrointestinal symptoms or disease severity.

In our case, the patient showed a difference in the detection of viral RNA after time, with a sudden positivity of nasopharyngeal swabs during the reinfection, despite the first infection. 

The complete pulmonary remission from the previous infection and the absence of any pulmonary abnormality in the second, both revealed by CT scan, indicate that the previous SARS-CoV-2 (RNA virus) infection may have resulted in a lower incidence of pneumonia or other neurological manifestations such as those attributed to a post-COVID-19 condition. One reason could be the acquisition of the so-called “mucosal immunity”, recently described as a key player in deciphering natural and vaccine-induced defenses [46]. From this perspective, a live attenuated oral vaccine could be considered as a possible strategic vaccine approach, as Albert Sabin did to eradicate poliomyelitis caused by poliovirus, an orofecal RNA virus that replicates in the intestine [47]. The stimulation and activity of mucosal surface IgA, not only in the respiratory tract but also in the oral and intestinal mucosa, and the activation of a prompt CD8 cell response proved to be of crucial importance in immediately counteracting a second reinfection [48]. Particularly with regard to post-acute COVID-19, live attenuated vaccines, virus-vectored vaccines and recombinant subunit vaccines are usually very effective, stimulating superficial mucosal immunity, and are often sufficient in a single dose to induce lasting immunity [49,50,51]. From this perspective, a nasal spray vaccine may also be useful [50].

In conclusion: This case showed a difference in the detection of viral RNA after time in the case of a reinfection in the same patient. Furthermore, in line with other studies [35,36], the presented case shows a reinfection with less severe symptoms, confirmed by radiological evidence with a healthy lung picture after 18 months, not affected by the second infection. We hypothesized that the symptoms in the reinfection were mild without lung involvement probably due to the stimulation and activity of mucosal immunity in the oral and intestinal mucosa that occurred in the previous infection.

## Figures and Tables

**Figure 1 medicina-58-00642-f001:**
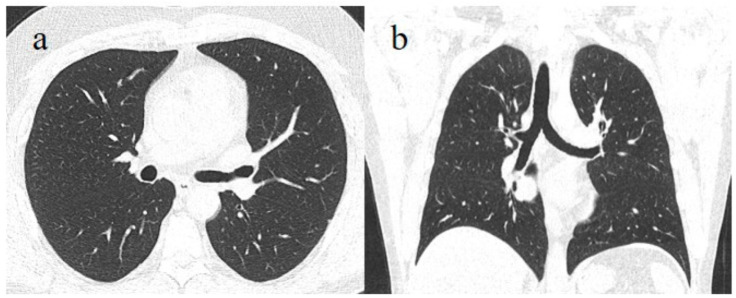
Chest CT showed absence of COVID-19 pneumonia with normal lung appearance on the (**a**) axial plane and (**b**) coronal plane.

**Table 1 medicina-58-00642-t001:** Main laboratory analyses result with the reporting systemic unit (SU) of measurements and the normal value range.

Laboratory Parameters	SU	Patient’s Value	Normal
Hemoglobin	mg/dL	16.2	13.0–17.5
Mean cell volume	fL	94.2	80.0–98.0
Platelets count	X1000/µL	167	140.00–450.00
White blood count	X1000/µL	6.29	4.00–11.0
Neutrophils	%	70.8	40.0–75.0
Lymphocytes	%	21.4	20.0–50.0
Monocytes	%	6.8	0.0–11.0
Eosinophils	%	0.8	0.0–0.7
Basophiles	%	0.2	0.0–0.2
Aspartate transaminases	U/L	25	<37
Alanine transaminases	U/L	30	<41
Glycemia	mg/dL	94	60–110
Creatinine	mg/dL	1.01	0.7–1.3
Cholinesterase	U/L	8850	4850–12,000
C Reactive Protein	mg/dL	0.22	<0.5
D-Dimer	mg/dL	193.4 ng/mL	<500

## Data Availability

Data available on request due to restrictions e.g., privacy or ethics.

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
