# Peer review of "SARS-CoV-2: Reinfection after 18 Months of a Previous Case with Multiple Negative Nasopharyngeal Swab Tests and Positive Fecal Molecular Test"

_medicina, 2022, doi:10.3390/medicina58050642_

Round 1
Reviewer 1 Report
The paper by Brogna et al. requires significant improvement as well as updated information as outlined below:
- The title is not correct as the fecal positivity was only present in the first illness.
- Lines 38-39: Limited evidence in literature on fecal-oral transmission for SARS-CoV-2 infection.
- Lines 50-51: The statement is incorrect as there are numerous data on reinfection especially with the Omicron variant.
- Line 60: To provide name of PCR test kit.
- Line 65: To provide more information on Abbott test kit.
- Line 105-106: The intranasal vaccination has been demonstrated to induce both the mucosal and systemic immune responses (Aqu Alu, Li Chen, Hong Lei, Yuquan Wei, Xiaohe Tian, Xiawei Wei, EBioMedicine. 2022 Feb; 76: 103841. Published online 2022 Jan 24. doi: 10.1016/j.ebiom.2022.103841)
- Line 109: The conclusion is incomplete and need to strengthen it more.
Author Response
Answer
We thank the Reviewer for the valuable comments and suggestions. All the modifications and corrections are marked in yellow. We hope that they will be found satisfactory.
The title is not correct as the fecal positivity was only present in the first illness.
,As suggested, we have reshaped the title to: “ SARS-CoV-2: Reinfection after 18 months of the previous case with multiple negative nasopharyngeal swab tests and positive faecal molecular test”.
Lines 38-39: Limited evidence in literature on fecal-oral transmission for SARS-CoV-2 infection.
R: we added other 10 references, and in accordance with the reviewer we replace the verd “ should be considered” with “ could be considered”. We have also added the sentence: and the possibility of fecal-oral transmission continue to be under investigations.
Lines 50-51: The statement is incorrect as there are numerous data on reinfection especially with the Omicron variant.
R: We apologize to the reviewer, and have reviewed the international literature more carefully. We have reworded the statement and added more indicative references. The line 50-51(now Line 53) are replaced in: “ Re-infection with SARS-CoV-2 in previously cured subjects is a phenomenon that is observed especially with the last B.1.1.529 (Omicron) variant ” ( 23-25)
Line 60: To provide name of PCR test kit.
We have added it in the case presentation. The patient also had thoracic pain. The first positive nasal antigenic swab was followed by confirmation of a positive molecular nasopharyngeal swab (Copan lot 2127827) with RT-PCR analysis (Viasure real time PCR detection kit-Cortest Biotec).
Line 65: To provide more information on Abbott test kit.
R: we added the information: Line 74 ( Abbott: AdviseDx SARS-CoV-2 IgG II assay - chemiluminescent microparticle immunoassay (CMIA))
Line 105-106: The intranasal vaccination has been demonstrated to induce both the mucosal and systemic immune responses (Aqu Alu, Li Chen, Hong Lei, Yuquan Wei, Xiaohe Tian, Xiawei Wei, EBioMedicine. 2022 Feb; 76: 103841. Published online 2022 Jan 24. doi: 10.1016/j.ebiom.2022.103841)
R: We fully agree with the reviewer. We did not want to talk about recently produced vaccines so as not to generate misunderstandings for or against a current method, but since the reviewer suggests it we insert the reference he suggested and rephrase the sentence as follows line 142-146. Especially with regard to post-acute COVID-19, live attenuated vaccines, virus-vectored vaccines, recombinant subunit vaccines, are usually very effective, stimulating superficial mucosal immunity, often in a single dose, are often sufficient to induce lasting immunity [49-51]. In this perspective also nasal spray vaccine may be useful [50].
Line 109: The conclusion is incomplete and need to strengthen it more.
R: We thank the reviewer and in line with his suggestion we completed the conclusion, within the word count limit for a short communication paper. The sentence has been reworded as follows, line 143-149: “In conclusion: This case showed a difference in the detection of viral RNA after time with reinfection in the same patient. Furthermore, in line with other studies [35,36], the presented case shows a reinfection with less severe symptoms, confirmed by ct-Scan radiological evidence with a healthy lung picture after 18 months and not affected by the second infection. We hypothesized that the symptoms in the reinfection were mild without lung involvement probably due to the stimulation and activity of mucosal immunity in the oral and intestinal mucosa that occurred in the previous infection”.

Reviewer 2 Report
Thank you for inviting me to review this case study. Authors have described a case having re-infection of COVID-19. There were diagnostic challenges for this patient when he was initially not diagnosed with multiple qRT-PCR tests (nasal swab) but later emerged as a positive case when the fecal sample was used. The authors have presented the case in a comprehensive manner. Their intention is to underscore the importance of fecal samples during the diagnosis of COVID-19. The suggestion to use both nasal/oral or fecal samples for the diagnosis of COVID-19 is not aggressively supported by the appropriate references. The authors have provided some references to similar cases at the end of the discussion. I suggest supporting the claim with more references and describing the sensitivity of nasal and fecal swabs. There are many re-infection cases reported in the literature, even among vaccinated individuals. How this case adds new knowledge to the existing literature, this point must be addressed in the introduction section of the manuscript. There is a need to provide information about the current status of the patient as outcome data is only available till day 10 of presentation to the health facility. The authors have described this case in reference no. 15, I suggest briefly describing the first infection in the discussion section so readers can get a quick glimpse of this patient during their first infection. What is the vaccination status of this patient? What are the other comorbid conditions in this patient? Demographic details of this patient are missing (I know it is somewhat present in reference 15 but please put some details in this manuscript too, as these parameters affect the outcomes of patients as well as their susceptibility for re-infection)
Author Response
Answer
We thank the Reviewer for the valuable comments and suggestions. All the modifications and corrections are marked in yellow. We hope that they will be found satisfactory.
There were diagnostic challenges for this patient when he was initially not diagnosed with multiple qRT-PCR tests (nasal swab) but later emerged as a positive case when the fecal sample was used. The authors have presented the case in a comprehensive manner. Their intention is to underscore the importance of fecal samples during the diagnosis of COVID-19. The suggestion to use both nasal/oral or fecal samples for the diagnosis of COVID-19 is not aggressively supported by the appropriate references. The authors have provided some references to similar cases at the end of the discussion. I suggest supporting the claim with more references and describing the sensitivity of nasal and fecal swabs.
R: We introduced in line 120 in the discussion the following sentence: Real-time quantitative reverse transcriptase-polymerase chain reaction is the gold-standard laboratory technique for the identification of SARS-CoV-2 in the clinical setting, and nasopharyngeal swabs are considered the primary sample for the COVID-19 test with a sensitivity around the 70-80% [39-41]. However, false negative results are also possible [39,42] and other specimens including fecal viral testing that has high specificity to detect SARSCoV-2 despite low sensitivity ranging from 37% to 60% have also been considered [42,43]. However, some studies demonstrated that the clearance time of COVID-19 in the digestive tract is later than that in the respiratory tract [44,45]. Recently, Wu et al [15] found that fecal positivity for SARS-CoV-2 was independent from the manifestation of gastrointestinal symptoms or disease severity.
We have also added similar studies of reinfections in the initial part of the discussion
This case is in line with other reports [35,36] and studies [26,27,34,37], which show that re-infected cases have a more attenuated and less severe clinical course than the first time
There are many re-infection cases reported in the literature, even among vaccinated individuals. How this case adds new knowledge to the existing literature, this point must be addressed in the introduction section of the manuscript.
R: At line 50-51 (now line 53) the sentence has been replaced in: “ Re-infection with SARS-CoV-2 in previously cured subjects is a phenomenon that is observed especially with the last B.1.1.529 (Omicron) variant ” ( 23-25) and the sentences in lines 58-66: In this brief supplement of previously published work [29], we report for the first time a difference in the detection of viral RNA in the same patient and how the pulmonary aspect radiologically differs between the first and second infections.
We previously noted that viral RNA was found in the faecal swab despite multiple negative nasopharyngeal molecular tests. Almost 18 months after the first episode, the same unvaccinated subject was re-infected. However, molecular testing from nasopharyngeal swab revealed viral load this time. In addition, while at chest CT, in the first episode, was visible COVID-19 typical pneumonia, after 18 months, it shows a normal lung, not affected by the second infection.
There is a need to provide information about the current status of the patient as outcome data is only available till day 10 of presentation to the health facility.
R: We have added at line 106-107 “The patient tested negative on day 10 and the clinical state of discomfort lasted 8 days in total. Currently, the patient has no symptoms and his antibody titer is 25563.3 AU/mL SARS-CoV-2 IgG (Abbott).
The authors have described this case in reference no. 15, I suggest briefly describing the first infection in the discussion section so readers can get a quick glimpse of this patient during their first infection. What is the vaccination status of this patient? What are the other comorbid conditions in this patient? Demographic details of this patient are missing (I know it is somewhat present in reference 15 but please put some details in this manuscript too, as these parameters affect the outcomes of patients as well as their susceptibility for re-infection)
We have added a summary of the case in the initial part of the discussion
Cases of re-infection are usually reported in the healthcare workers and the rate of reinfection in the general population is underestimate [34]. SARS-CoV-2 re-infection of the previously described case [29] showed milder symptoms. In the first infection the patient showed, despite the multiple nasal swabs negativity for SARS-CoV-2 with fecal test positivity, also sign of COVID-19 pneumonia on CT with IgM positivity on serology. However, the patient didn’t show any lung involvement during the second infection.
.
We have added the description of the first paper ( now reference n 29) to lines 68-73: “The 44-year-old patient, Caucasian, with arterial hypertension disease, chronically taking ace-inhibitors, non-smoker, of good socio-economic conditions, had his first Covid-19 infection about 18 months earlier. He came the last time, to our observation with severe dyspnea, desaturation, and heart palpitations. The first time we found positive SARS-CoV-2 molecular tests in his faeces, despite six negative nasopharyngeal molecular swabs. At the end of January 2022 (i.e., almost 18 months after the infection reported in [29]), he was re-infected by SARS-CoV-2 and he was not vaccinated for SARS-CoV-2.
We thank the reviewer very much for his comments and for improving the work.
Submission Date
05 April 2022

Reviewer 3 Report
Title: SARS-CoV-2: a case of reinfection with multiple negative naso-2 pharyngeal swab tests and faecal molecular test positivity 18 months later.
General Comment: The authors describe the reinfection of the same patient who was previously infected with SARS-CoV-2, after nearly 18 months. The patient in the previous infection showed negative multiple qRT-PCR results by nasal SARS-CoV-2 but positive results on faecal sample. Despite a high natural antibody titer, the patient was affected by SARS-CoV-2 after 18 months and the nasal swab showed readily positive results. The clinical course appeared to be more attenuated and the radiographic examinations performed showed no signs of pulmonary involvement. This manuscript is an important contribution as the authors have also put forward an interesting approach to an oral vaccine for COVID-19, e.g. orofaecal RNA virus would replicate in the intestine.
The study has sound methodology.
The analysis has been described well.
I have a minor comment:
- Lines 24 and 53: Please correct “Sars-Cov-2” to SARS-CoV-2.
- Discussion on Muscosal immunity and less severe symptoms of COVID-19 can be elaborated.
Author Response
Answer:
We thank the reviewer very much for appreciating the work and improving it.
Lines 24 and 53: Please correct “Sars-Cov-2” to SARS-CoV-2.
R: Done
Discussion on Mucosal immunity and less severe symptoms of COVID-19 can be elaborated.
R: we have included the required commentary on the lines 140-143.: “The stimulation and activity of mucosal surface IgA not only in the respiratory tract but also in the oral and intestinal mucosa and the activation of a prompt CD8 cell response proved to be of crucial importance in immediately counteracting a second re-infection [48]”
